# Sorafenib with Transarterial Chemoembolization Achieves Improved Survival vs. Sorafenib Alone in Advanced Hepatocellular Carcinoma: A Nationwide Population-Based Cohort Study

**DOI:** 10.3390/cancers11070985

**Published:** 2019-07-15

**Authors:** Victor C. Kok, Yu-Ching Chen, Yang-Yuan Chen, Yu-Chieh Su, Ming-Chang Ku, Jung-Tsung Kuo, Go J. Yoshida

**Affiliations:** 1Division of Medical Oncology, Department of Internal Medicine, Kuang Tien General Hospital, Taichung 43303, Taiwan; 2Disease Informatics Research Group, Department of Bioinformatics and Medical Engineering, Asia University Taiwan, Taichung 41354, Taiwan; 3Cancer Biology and Therapeutics: High-Impact Cancer Research Postgraduate Certificate Program, Harvard Medical School, Boston, MA 02115, USA; 4Department of Gastroenterology, Changhua Christian Medical Foundation Changhua Christian Hospital, Changhua 50006, Taiwan; 5Faculty of Medicine, College of Medicine, Kaohsiung Medical University, Kaohsiung 80708, Taiwan; 6Division of Hematology and Oncology, Department of Internal Medicine, Kaohsiung Medical University Hospital, Kaohsiung 80708, Taiwan; 7Interventional Radiology Unit, Department of Radiology, Kuang Tien General Hospital, Taichung 43303, Taiwan; 8Artificial Intelligence Center for Medical Diagnosis, China Medical University Hospital, Taichung 40447, Taiwan; 9Department of Pathology and Oncology, Juntendo University School of Medicine, Tokyo 113-8421, Japan; 10Faculty of Medical Science, Graduate School Juntendo University, Tokyo 113-8421, Japan

**Keywords:** sorafenib, hepatocellular carcinoma, transarterial chemoembolization, add-on therapy, propensity analysis

## Abstract

We hypothesized that sorafenib plus transarterial chemoembolization (TACE) would confer survival benefits over sorafenib alone for advanced hepatocellular carcinoma (aHCC). We investigated this while using the population-based All-Cancer Dataset to assemble a cohort (*n* = 3674; median age, 60; 83% men) of patients receiving sorafenib for aHCC (Child-Pugh A) with macro-vascular invasion or nodal/distant metastases. The patients were classified into the sorafenib-TACE group (*n* = 426) or the propensity score-matched sorafenib-alone group (*n* = 1686). All of the participants were followed up until death or the end of the study. Time-dependent Cox model and the Mantel–Byar test were used for survival analysis. During the median follow-ups of 221 and 133 days for the sorafenib-TACE and sorafenib-alone groups, 164 (39%) and 916 (54%) deaths occurred, respectively; the corresponding median overall survivals (OS) were 381 and 204 days, respectively (hazard ratio, HR: 0.74; 95% confidence interval, CI, 0.63–0.88; *p* = 0.021). The one-year and six-month OS were 53.5% and 80.3% in the sorafenib-TACE group and 32.4% and 54.4% in the sorafenib-alone group, respectively. The major complications were comparable between the two groups. The addition of TACE to sorafenib improves survival, with a 26% reduction in mortality. These findings provide strong real-world evidence that supports this combination strategy for eligible Child-Pugh A aHCC patients.

## 1. Introduction

The global cancer statistics from 2018 (GLOBOCAN 2018) estimated that there were 841,080 cases of primary liver cancer worldwide, resulting in 781,631 deaths [1]. The treatment options are limited for advanced stage hepatocellular carcinomas (aHCC), described as Barcelona Clinic Liver Cancer (BCLC) stage C, or designated as T4 or N1 or M1 in the 2017-Tumor-Node-Metastasis staging system, which includes single or multiple hepatocellular carcinomas of any size involving a major branch of the portal vein or hepatic vein, or having extrahepatic lymph nodes or distant metastases [2]. In a highly endemic region, the majority (53.6%) of hepatocellular carcinoma (HCC) cases were staged as BCLC stage C, at diagnosis, because early stages of HCC were asymptomatic [3].

The Sorafenib Hepatocellular Carcinoma Assessment Randomized Protocol (SHARP) trial [4] and the Asia-Pacific parallel study [5] have demonstrated that the antiproliferative and antiangiogenic molecularly-targeted agent, sorafenib, which is an oral multi-kinase inhibitor, improves the overall survival (OS) in patients with aHCC when it was administered at a dose of 400 mg twice a day, when compared with a placebo. The median OS in the SHARP trial was 10.7 and 7.9 months in the sorafenib and placebo groups, respectively, with a hazard ratio (HR) of 0.69 [4]. In the Asia-Pacific parallel trial, the median OS was 6.5 months in the sorafenib group, when compared with 4.2 months in the placebo group, with a similar HR at 0.68 [5]. The different patient demographics may explain the shorter OS reported in the Asia-Pacific trial; there were more patients with poorer performance status, a higher number of hepatic tumors, extrahepatic metastases, and higher serum concentrations of alpha-fetoprotein, when compared to patients in the SHARP trial. Sorafenib has become the first-line standard treatment for aHCC since these two trials reported positive survival outcomes, specifically for BCLC stage C disease.

During the sorafenib era, investigators worldwide began to investigate whether transarterial chemoembolization (TACE) could improve survival in patients receiving sorafenib for aHCC [6,7,8,9,10,11,12,13,14,15,16,17]. There are few comparative trials [6,13,18], and even fewer prospective clinical trials [6], which directly compared the effectiveness of sorafenib with TACE vs. sorafenib alone among patients with aHCC. The only prospective multi-center phase III trial comprising a total of 339 South Korean patients with unresectable HCC (including 25% with either BCLC Stage A, or B) reported that TACE did not improve OS when it was added to sorafenib. The median OS for the sorafenib with TACE arm was 12.8 months vs. 10.8 months in the sorafenib alone arm (HR: 0.91; 90% CI 0.69–1.21; *p* = 0.290) [6]. It is still unclear whether TACE added to sorafenib targeted treatment confers improved survival among the patients with aHCC.

The rationale for incorporating TACE during sorafenib treatment results from the clinical observation that lipiodol deposition is significantly increased when sorafenib is added after TACE [19]. Certain local tumors can be controlled with TACE, thus preventing local disease progression. A large population-based study could help to provide long overdue, real-world evidence on the issue [20,21]. Therefore, we conducted a nationwide cohort study while using propensity score matching algorithms to examine whether TACE add-on therapy would confer survival benefit.

## 2. Results

### 2.1. Demographics of the Two Groups after Propensity Score Matching (PSM)

The final cohort comprised 2112 patients from1 August 2012, to 31 December 2013 (Figure 1). Baseline demographics and disease characteristics, including lymph node and distant organ metastases status after PSM, were generally balanced between the two treatment groups (Table 1). The median age of this cohort was 60 years; 83% were men. Approximately 94% of the patients had a Charlson–Deyo comorbidity index of 3 or higher, which indicated more co-existing medical comorbidities. Seventy-five percent of the patients were treated at a tertiary (highest-ranking) medical center (institution). More importantly, cancer extent by metastatic site was well balanced between the sorafenib with TACE and sorafenib alone treatment groups, in terms of lymph node involvement (6.8% vs. 6.5%), pulmonary metastases (21.1% vs. 20.2%), adrenal gland metastasis (2.1% vs. 2.2%), osseous metastases (12.9% vs. 12.5%), and peritoneal metastases (3.3% in either group) (Table 1). The metformin users were balanced between the two treatment groups, comprising approximately 14% of all the patients [22,23,24].

### 2.2. Follow-Up and Survival Analysis Results

There were 164, and 916 deaths in each arm, respectively, with a median follow-up time of 221 days (quartile, 140–345 days) for the sorafenib + TACE add-on arm, and 133 days (quartile, 68–251 days) for the sorafenib alone arm. The median OS was 381 days (95% CI, 327–435 days) (median: 12.5 months; 95% CI, 10.8–14.3 months) for the sorafenib + TACE arm, which is statistically significantly longer than the sorafenib alone arm (median: 204 days, 95% CI, 188–221 days; median 6.7 months, 95% CI, 6.2–7.3 months) with the reduction of the risk of death by 26% (HR: 0.74; 95% CI, 0.63–0.88, time-dependent Cox) (Table 2). Kaplan–Meier survival curves were compared while using the Mantel-Byar test (χ^2^ = 6.10, *p* = 0.013) (Figure 2).

Patients in the sorafenib + TACE group had improved six-month (80.3% vs. 54.4%) and one-year survival rates (53.5% vs. 32.4%) than the patients receiving sorafenib alone. The median time to progression (equivalent to sorafenib discontinuation) was 144 days (95% CI, 127–161) (median: 4.7 months; 95% CI, 4.2–5.3) for sorafenib + TACE group, in contrast to 86 days (95% CI, 80–92) (median: 2.8 months; 95% CI, 2.6–3.0) in the sorafenib alone group, which represents a 24% reduction in risk (HR: 0.76; 95% CI, 0.65–0.89) (Table 2).

### 2.3. Subgroup Analysis

Subgroup analysis that was based on the characteristics of the studied patients with aHCC receiving sorafenib with or without additional TACE revealed favorable outcomes for sorafenib + TACE. Cox regression modeling was used to test for heterogeneity in the subgroup analysis. Female sex was associated with a statistically non-significant reduction of risk (HR: 0.74, 95% CI, 0.47–1.17), and it had fewer patient numbers (*n* = 347). A similar explanation could be for the Charlson comorbidity index ≤ 2 subgroup (HR: 0.61; 95% CI, 0.20–1.82), which comprised 135 patients. Likewise, the metformin users (*n* = 299) had an HR of 0.66 (95% CI, 0.40–1.11), favoring sorafenib + TACE. Patients presenting non-metastatic advanced HCC with vascular invasion (*n* = 1136) showed a trend towards significant risk reduction with an HR of 0.78 (95% CI, 0.59–1.02) (Figure 3). Figure 3 also demonstrates that the prevalence of bone metastasis in the current cohort was 12.6% (266/2112). The proportion of HBV infection-related HCC accounted for 34.7% (733/2112) in the post-matching cohort; whereas HCV-related, alcoholism-related, and other uninfected etiology HCC accounted for 65.3% of the entire cohort.

### 2.4. Major Complications Compared between Two Groups

Furthermore, the selected major complications, including HCC rupture, spontaneous bacterial peritonitis, esophageal variceal bleeding, hepatic encephalopathy, hepatic failure, and disseminated intravascular coagulation, were similar between the groups (Table 3).

### 2.5. Sensitivity Analysis of Survival Outcome

A similar trend was identified in a subsequent sensitivity analysis of survival outcome stratified by the number of additional TACE treatments a patient received (once vs. two or more sessions). Only one additional TACE treatment markedly improved the median OS (10.4 months; 95% CI, 9.6–11.1 months) as compared to the sorafenib-alone group. In the group of patients who received two or more TACE treatments, the adjusted HR improved to 0.41 (95% CI, 0.29 to 0.56) (Table 4).

## 3. Discussion

To our knowledge, this is the largest population-based cohort study that demonstrated a positive survival impact of adding TACE to sorafenib therapy, among propensity-score-matched patients with aHCC. The sorafenib-alone control group in our study had a comparable median OS to that of the Asia-Pacific parallel randomized trial (6.7 vs. 6.5 months) [5]. The median age of the current study cohort was numerically higher than that of the Asia-Pacific trial (60.4 vs. 51 years), with extrahepatic metastatic disease being present in 68.7%, and 46.2% of the patients in the Asia-Pacific trial, and this study, respectively. Our cohort of 2112 patients with aHCC demonstrated homogeneity in stage (BCLC stage C) and liver function (Child-Pugh class A).

When comparing the current results to that of comparative trials, other than the Asia-Pacific trial may not be meaningful. Two relevant South Korean studies had different populations compared to ours [6,13]. Approximately 25% patients of the entire cohort in the prospective randomized trial by Park et al. [6] had lower stage HCC (either BCLC A, or B), and only 35.7% had extrahepatic metastatic disease. The retrospective single-center analysis that was conducted by Choi et al. comprised younger patients (median age: 52, 95% CI, 26–75) [13].

In this real-world study, we minimized the risk of selection bias by using propensity score matching. This was used to create a counterfactual group after adequately employing a predicted probability of group membership (sorafenib + TACE treatment vs. sorafenib alone control), which was obtained from logistic regression, based on pre-selected demographic, and clinical characteristics. It was ascertained beforehand that patients in the sorafenib alone group did not undergo any TACE add-on treatment; therefore, this study is not subject to allocation bias. Mantel–Byar testing was used in the survival analysis to eliminate the risk of immortal time bias.

We believe that the modern theory of the potential synergistic effects of sorafenib in combination with TACE add-on treatment is still evolving, particularly on the molecular level. A prospective clinical study has identified that TACE treatment may result in increased serum vascular endothelial growth factor levels one to two days following the procedure, which could increase the chance of extrahepatic spread and unfavorable outcomes [25,26]. Previous studies have also suggested that sorafenib treatment may improve lipiodol retention in HCC [19]. The latter provides support of better local control of aHCC, which may improve the survival, even in a patient with extrahepatic metastatic disease. Our subgroup analysis lends support for this concept, since our data demonstrate that patients with aHCC with either solely bone metastases or any other sites of metastatic disease will benefit from sorafenib + TACE (HR: 0.63; 95% CI, 0.41–0.96 for bone-only metastasis; and, HR: 0.69; 95% CI, 0.55–0.86 for any site metastasis).

In patients with aHCC and Child-Pugh class A hepatic function reserve, local therapeutic options, like TACE, may be a treatment of choice, aiming to reduce tumor burden and hamper disease progression. In the lack of mature evidence, oncologists often do not offer TACE to patients with BCLC stage C aHCC in clinical practice, even if embolizable hepatic tumors are present. Our findings may provide another perspective, supporting the practice of adding TACE to sorafenib therapy. Tolerability to simultaneous sorafenib and TACE treatments is good, as major adverse events and complications between the sorafenib alone and the combined treatment groups were comparable.

There are several limitations to this study. Firstly, regorafenib might be used in a small percentage of patients after sorafenib failure [27]. However, as the drug is not covered by the national insurance, we could not track its use in the All-cancer dataset. During the study period, neither lenvatinib, nor immune checkpoint inhibitors had been approved for use in patients with aHCC in this country. Secondly, drug compliance, which may potentially affect the treatment outcome, could not be assessed in patients on sorafenib therapy.

## 4. Materials and Methods

We conducted a nationwide, population-based propensity score-matched cohort study to investigate whether the addition of TACE with sorafenib improves OS, when compared to sorafenib alone in patients with Child-Pugh A advanced stage HCC. The Kuang Tien General Hospital institutional review board approved the study (KTGH-10458). The requirement for informed consent was waived. The study is registered with ResearchRegistry (http://www.researchregistry.com/browse-the-registry.html#home/), and the unique identifying number is “researchregistry2017”. We reported the research results according to the STROBE guidelines for cohort studies (Appendix A).

### 4.1. Source of Data

The cancer dataset (Cd) constructed from the Taiwan National Health Insurance Research Database (NHIRD) is provided to scientists for research purposes after formal application in Taiwan. The NHIRD is an extensive computerized database, which is derived from the national health insurance system, containing registration files and original claim data for reimbursement maintained by the National Health Research Institutes, Taiwan. As of 2014, 99.9% of Taiwan’s population were enrolled in the single-payer national health insurance [28]. Data that matched any of the cancer-related diagnoses, and International Classification of Diseases, 9th Revision, Clinical Modification (ICD-9-CM) treatment or procedural codes, or other cancer-related codes were selected for constructing the Cd dataset. This included those ICD-9-CM codes with the first three digits between 140 and 239, and cancer-surgery-related codes with the first three digits between V57 and V58. The investigators in this study are familiar with the NHIRD clinical cancer studies [29,30,31,32]. A total of 81,983 patients with hepatocellular carcinoma were in the Cd between April 1, 2010 and December 31, 2013. The age-standardized annual incidence of primary invasive liver cancer in Taiwan in 2012 was 35.02 per 100,000 people [33]. Another group successfully carried out previous research on HCC while using the NHIRD [34].

### 4.2. Inclusion and Exclusion Criteria

HCC diagnosis was defined according to the major discharge diagnoses comprising ICD-9 codes 155.0. Validation was spontaneous and complete, because only those receiving sorafenib under the national health insurance preauthorization clearance were selected. Child-Pugh class A denotes a score of either 5 or 6, which indicated compensated liver reserve [35]. The inclusion criteria were as follows: advanced unresectable hepatocellular carcinoma with either extrahepatic lymph nodes or distant organ metastasis, large blood vessel invasion (i.e., a tumor invading either the main portal vein or the first branch of the left or right hepatic vein), or those with a liver function reserve of Child-Pugh class A with a score either 5 or 6. The exclusion criteria were as follows: Child-Pugh class B or C. This study excluded the patients with Barcelona Clinic Liver Cancer stages A, B, or D.

### 4.3. Patient Accrual and Cohort Assembly

A cohort of 72,828 patients with HCC was formed after excluding patients with concurrent cholangiocarcinoma (*n* = 8642), concurrent renal cell carcinoma (*n* = 512), and those with unknown sex (*n* = 1). Starting from August 1, 2012, sorafenib was reimbursed for patients with unresectable advanced hepatocellular carcinoma with either metastases or invasion of the portal vein or the first branch of either hepatic vein (RHV or LHV) by tumor; these patients had Child-Pugh class A liver function reserve. A strict preauthorization expert review process was carried out by medical oncologist reviewers for each case, before sorafenib administration. We retrieved data for all patients with HCC who received sorafenib (*n* = 3879). Among them, 205 patients were further excluded due to a follow-up duration of seven days or less. The patients who received TACE add-on therapy were assigned to the sorafenib with the TACE group (*n* = 426), with the first day of sorafenib treatment set as the index date. We used a propensity score matching algorithm to assemble the comparison group treated with sorafenib alone (*n* = 1686) (Figure 1). Cross-over to the other arm was not allowed. A tumor response assessment was performed every two months, as required for continuing sorafenib only in cases achieving complete remission, partial remission, or stable disease.

### 4.4. Propensity Score Matching

Propensity score analysis was based on guidelines that were proposed by a group from Duke University [36]. Propensity score analysis is useful in addressing the confounding factors that are inherent in observational studies, particularly the bias that arises from the apparent difference in outcome between two groups with different units of observation. The propensity score matching (PSM) was used; logistic regression was used to estimate the propensity score. Age, sex, Charlson-Deyo comorbidity index score, insurance premium category (socioeconomic surrogate), hospital volume, metastatic site, type 2 diabetes, metformin use, and chronic kidney disease were the variables included in the propensity score model (Table 1). Metformin use was included for PSM, because recent studies have demonstrated the potential beneficial effects of combining metformin and sorafenib on inhibiting the proliferation of HCC [22,23,24]. The matching algorithm and distance metric for the PSM was greedy matching with 0.2 caliper distance, and sampling without replacement. There were no missing data in any of the variables for propensity score estimation. The comparability of baseline characteristics in the matched groups was assessed while using the standardized mean difference.

### 4.5. Outcome Measures

All of the patients were followed until death, or last mentioned in the dataset. Drop-out was unlikely due to complete geographic coverage by the universal insurance system. OS, six-month OS, and one-year OS were calculated. The endpoint of time to sorafenib discontinuation was defined as the last day of sorafenib administration (i.e., date of death, date of disease progression) in the medication data file, which was used as a proxy for time to progression (TTP). All of the patients were tracked for major events or complications, including HCC rupture, spontaneous bacterial peritonitis, esophageal variceal bleeding, hepatic encephalopathy, hepatic failure, and disseminated intravascular coagulopathy.

### 4.6. Statistical Analysis

The use of propensity scores involves constructing a multivariate model to predict receipt of the add-on TACE treatment. A propensity score is a predicted probability of receiving TACE, given baseline covariates. The covariates are approximately equally distributed within subgroups that were defined by the propensity score. We estimated the propensity to receive add-on TACE with the use of predicted probabilities from a logistic-regression model. The Kaplan–Meier method was used for survival estimation. Extended Cox modeling was used to incorporate time-varying coefficients (TACE) since an apparent immortal time bias existed in the add-on therapy group, and the proportional hazards assumption could not hold [37]. To avoid the immortal time bias, comparison of the primary endpoint (OS) between the TACE add-on and no TACE groups was performed with the use of Mantel–Byar testing [38,39,40]. We estimated the effect size by calculating hazard ratios while using Cox regression modeling with time-varying covariates, because exposure (TACE) changed over time, with varying transition time among patients. Two-by-two Chi-square testing was used to calculate the odds ratios, and 95% confidence intervals (CI) were estimated. The *p*-values are two-sided and not adjusted for multiple testing. Statistical significance was set at *p* < 0.05 in all analyses. Sensitivity analysis was performed, and a forest plot was used to show the hazard ratio, and corresponding 95% CI derived from the Cox model of the following patient characteristics: age, sex, hospital volume, metastatic site, metformin use, CCI score, diabetes mellitus, chronic kidney disease, and etiology of HCC. The statistical analyses were performed while using the IBM SPSS v.22 software package (IBM Corp., Armonk, NY, USA).

## 5. Conclusions

Adding TACE to sorafenib achieves improved the OS, with a 26% reduction in mortality. To our knowledge, this is the largest population-based, proper person-time propensity analysis of real-world patients with aHCC, having Child-Pugh class A liver function reserve; the findings demonstrate a strong survival impact of adding TACE to sorafenib. These findings have the potential to influence clinical practice and to develop treatment strategies for patients with BCLC stage C aHCC. Sorafenib with additional TACE should be the treatment of choice for eligible patients. These findings will also encourage future prospective randomized trials and the evaluation of biomarkers for treatment efficacy.

## Figures and Tables

**Figure 1 cancers-11-00985-f001:**
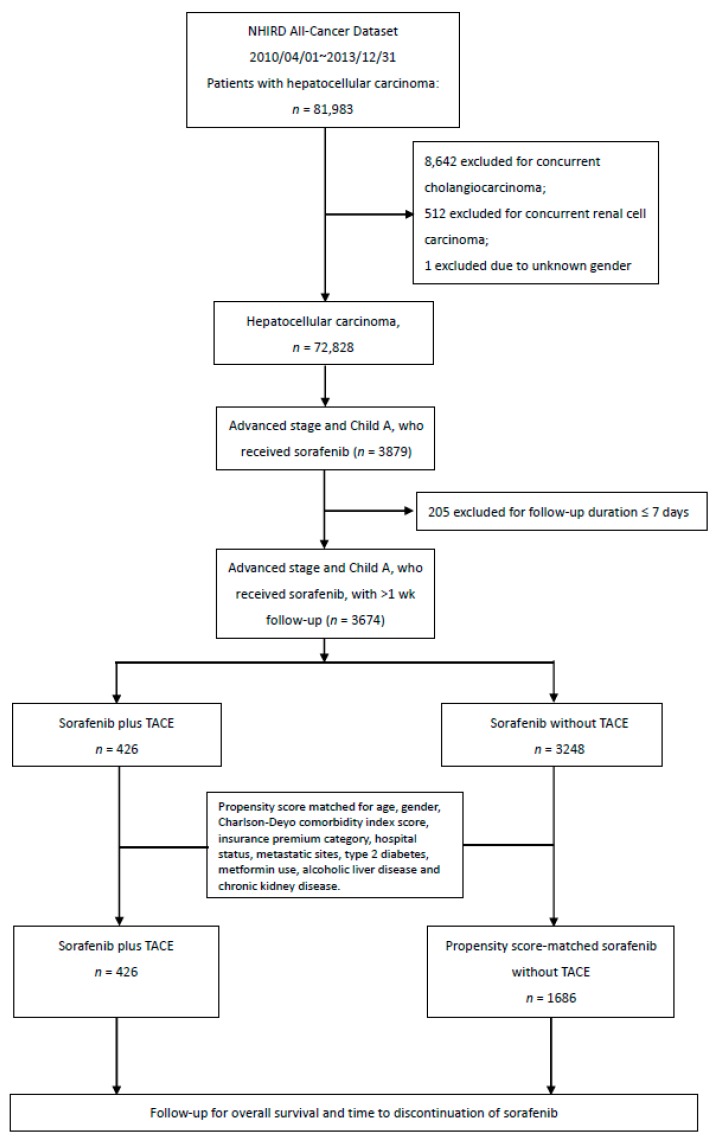
Consort diagram of the study flow towards the final cohort of 2112 patients.

**Figure 2 cancers-11-00985-f002:**
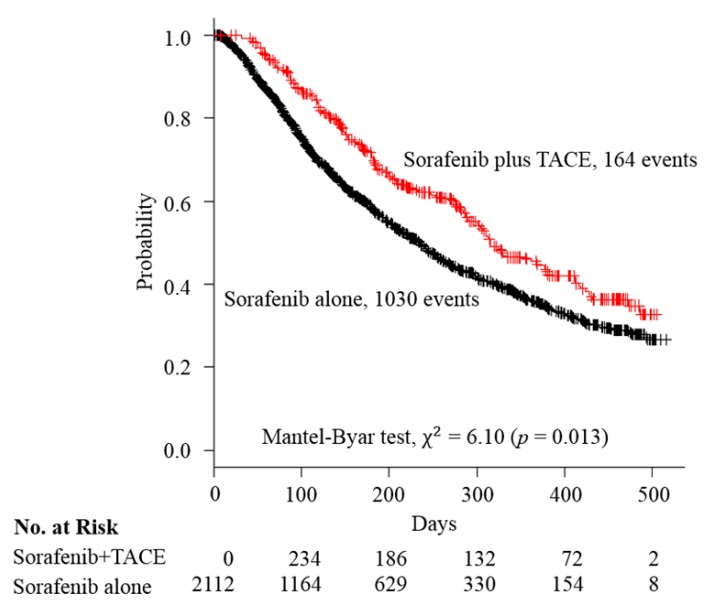
Kaplan–Meier curves of overall survival compared between the sorafenib + TACE arm and sorafenib alone arm. Abbreviation: TACE, transcatheter arterial embolization.

**Figure 3 cancers-11-00985-f003:**
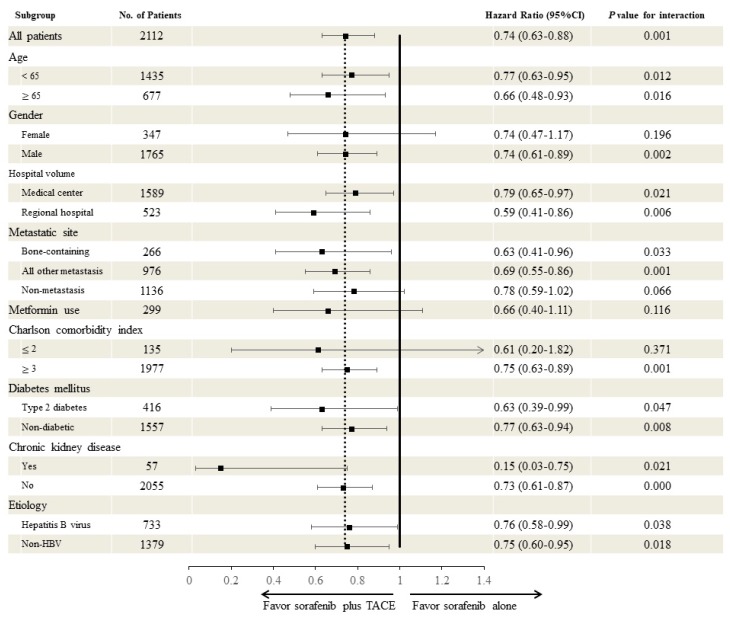
Subgroup analysis based on the characteristics of the cohort of patients with unresectable hepatocellular carcinoma receiving sorafenib with or without add-on TACE. Time-dependent Cox regression modeling was performed to test for heterogeneity in the subgroup analysis. 139 subjects had unknown diabetes mellitus status. Abbreviations: TACE, transcatheter arterial embolization.

**Table 1 cancers-11-00985-t001:** Baseline characteristics of patients with advanced hepatocellular carcinoma (Child-Pugh A) receiving sorafenib with or without add-on transarterial chemoembolization (TACE), before and after propensity-score matching. (All-Cancer Dataset 2008–2010).

Characteristics	Before Propensity-Score Matching				After Propensity-Score Matching			
	Without TACE (*n* = 3248; 88.4%)	With TACE (*n* = 426; 11.6%)	Standardized Mean Difference	*p*-Value	Without TACE (*n* = 1686; 79.8%)	With TACE (*n* = 426; 20.2%)	Standardized Mean Difference	*p*-Value
Age								
Mean (Median)	63.0 (63.1)	59.3 (60.4)	−0.305	0.000	60.0 (60.0)	59.3 (60.4)	−0.023	0.277
IQR	55.2–72.5	50.7–68.7			51.8–67.8	50.7–68.7		
Gender, *n* (%)								
Male	2528 (77.8)	355 (83.3)	0.147	0.009	1410 (83.6)	355 (83.3)	−0.013	0.883
Female	720 (22.2)	71 (16.7)			276 (16.4)	71 (16.7)		
CCI, *n* (%)								
0	53 (1.6)	0 (0.0)	0.395	0.000	3 (0.2)	0 (0.0)	0.007	0.682
1–2	413 (12.7)	27 (6.3)			105 (6.2)	27 (6.3)		
≥3	2782 (85.7)	399 (93.7)			1578 (93.6)	399 (93.7)		
Insurance premium category								
Lowest	429 (13.2)	53 (12.4)	0.003	0.357	194 (11.5)	53 (12.4)	−0.015	0.320
Low	1121 (34.5)	146 (34.3)			515 (30.5)	146 (34.3)		
Medium	931 (28.7)	122 (28.6)			584 (34.6)	122 (28.6)		
High	79 (2.4)	16 (3.8)			54 (3.2)	16 (3.8)		
Highest	196 (6.0)	33 (7.7)			123 (7.3)	33 (7.7)		
Hospital status								
Academic center	2209 (68.0)	321 (75.4)	0.170	0.002	1268 (75.2)	321 (75.4)	−0.03	0.951
Non-academic center	1039 (32.0)	105 (24.6)			418 (24.8)	105 (24.6)		
Metastasis								
Lymph nodes	174 (5.4)	29 (6.8)	0.058	0.218	110 (6.5)	29 (6.8)	0.006	0.833
Lungs	599 (18.4)	90 (21.1)	0.066	0.182	340 (20.2)	90 (21.1)	0.019	0.660
Adrenal gland	59 (1.8)	9 (2.1)	0.021	0.670	37 (2.2)	9 (2.1)	−0.004	0.918
Bone	341 (10.5)	55 (12.9)	0.072	0.131	211 (12.5)	55 (12.9)	0.003	0.826
Peritoneum	82 (2.5)	14 (3.3)	0.043	0.354	55 (3.3)	14 (3.3)	−0.016	0.980
Type 2 diabetes								
Yes	534 (16.4)	84 (19.7)	0.082	0.089	332 (19.7)	84 (19.7)	−0.002	0.990
No	2909 (84.4)	350 (80.6)			1354 (80.3)	342 (80.3)		
Metformin use								
Yes	407 (12.5)	60 (14.1)	0.045	0.365	239 (14.2)	60 (14.1)	−0.006	0.962
No	3034 (88.1)	374 (86.2)			1447 (85.8)	366 (85.9)		
Alcoholic liver disease								
Yes	128 (3.9)	26 (6.1)	0.090	0.036	97 (5.8)	26 (6.0)	0.005	0.783
No	3120 (96.1)	400 (93.9)			1589 (94.2)	400 (93.9)		
Chronic kidney disease								
Yes	109 (3.4)	12 (2.8)	−0.033	0.558	45 (2.7)	12 (2.8)	0.011	0.866
No	3139 (96.6)	414 (97.2)			1641 (97.3)	414 (97.2)		

Propensity score 0.13 (0.04) 0.13 (0.05) *p* = 0.400.

**Table 2 cancers-11-00985-t002:** Efficacy outcomes stratified by sub-cohorts with or without TACE after propensity score matching.

Outcome Measures	Sorafenib + TACE *n* = 426	Sorafenib Alone *n* = 1686	Hazard Ratio (95% Confidence Interval, CI)	*p*-Value
Median follow-up (Quartile) days	221 (140–345)	133 (68–251)	-	-
Outcome N (%) Deaths	164 (38.5)	916 (54.3)	-	0.000
Median overall survival (OS) in days (95% CI)	381 (327–435)	204 (188–221)	0.74 * (0.63–0.88)	0.021
Median overall survival (OS) in months (95% CI)	12.5 (10.8–14.3)	6.7 (6.2–7.3)		
6-month OS	80.3%	54.4%	-	-
1-year OS	53.5%	32.4%	-	-
Median time (days) to sorafenib discontinuation (95% CI)	144 (127–161)	86 (80–92)	0.76 (0.65–0.89)	0.001
Median time (mo.) to sorafenib discontinuation (95% CI)	4.7 (4.2–5.3)	2.8 (2.6–3.0)		

SD: standard deviation; TACE: Trans-Arterial Chemo-Embolization. * Cox regression model with time-varying covariate because exposure (i.e., TACE) changed over time with varying transition period among patients. Adjusted HR by age, sex, medical comorbidity severity, and socioeconomic state, hospital volume status, metastasis to lymph nodes, lungs, adrenal gland, bone and peritoneum, type 2 diabetes, metformin use, alcoholic liver disease, and chronic kidney disease.

**Table 3 cancers-11-00985-t003:** Major complications or adverse events.

Major Event	Sorafenib + TACE Event(s) (SD)	Sorafenib Alone Event(s) (SD)	Odds Ratio ^†^ (95% CI)	*p*-Value
Rupture of hepatocellular carcinoma	9 (2.1)	34 (2.0)	1.05 (0.50–2.20)	0.900
Spontaneous bacterial peritonitis	11 (2.6)	76 (4.5)	0.56 (0.30–1.07)	0.74
Esophageal variceal hemorrhage	9 (2.1)	28 (1.7)	1.28 (0.60–2.73)	0.525
Hepatic encephalopathy	48 (11.3)	251 (14.9)	0.73 (0.52–1.01)	0.056
Hepatic failure	7 (1.6)	12 (0.7)	2.33 (0.91–5.96)	0.069
Disseminated intravascular coagulopathy	1 (0.2)	4 (0.2)	0.99 (0.11–8.88)	0.992

^†^ Chi-squared test (2 × 2). Abbreviations: CI, confidence interval; SD, standard deviation; TACE, transarterial chemoembolization.

**Table 4 cancers-11-00985-t004:** Survival outcome of sub-cohorts stratified by the number session of additional transarterial chemoembolization (TACE), administered once-only versus two or more sessions, when compared with the sorafenib-alone group.

Measures	Sorafenib Alone *n* = 1686 (Reference)	Sorafenib + TACE × 1 *n* = 293		Sorafenib + TACE ≥ 2 *n* = 133	
Median follow-up (Quartile) days	133 (68–251)	196 (117–303)		313 (195–420)	
Outcome N (%) Deaths	916 (54.3)	125 (42.7)		39 (29.3)	
Median overall survival in days (95% CI)	204 (188–221)	315 (292–338)	**Adjusted HR *** = 0.97 (95% CI, 0.80–1.18)	NR	**Adjusted HR *** = 0.41 (95% CI, 0.29–0.56)
Median overall survival in months (95% CI)	6.7 (6.2–7.3)	10.4 (9.6–11.1)		NR	

CI: confidence interval; HR: hazard ratio; NR: not reached. * Cox regression model with time-varying covariate because exposure (i.e., TACE) changed over time with varying transition period among patients. Adjusted HR by age, sex, medical comorbidity severity, and socioeconomic state, hospital volume status, metastasis to lymph nodes, lungs, adrenal gland, bone and peritoneum, type 2 diabetes, metformin use, alcoholic liver disease, and chronic kidney disease.

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
