# Peer review of "Sorafenib with Transarterial Chemoembolization Achieves Improved Survival vs. Sorafenib Alone in Advanced Hepatocellular Carcinoma: A Nationwide Population-Based Cohort Study"

_cancers, 2019, doi:10.3390/cancers11070985_

Round 1
Reviewer 1 Report
Dear Authors,
Congratulations on the collection so big material.
My notes are as follows:
Study group comes from Taiwan.This region differs in the etiology of cancer in relation to other parts of the world. There is no information about possible etiology in the manuscript. Only in Table 1 alcoholic liver diseases cases are shown. There is no data about HBV or HCV infections and antiviral therapy during cancer treatment. In so big Taiwan DataBase such informations should exist, especially that inFfigure 3 you put the etiology of HBV or non-HBV.
In Table 4 presented data concern patients with sorafenib and TACE treatment. Selected group of patients who received two or more TACE treatment is presented. I could not find out if the next TACE was done because of HCC progression or was planned from the beginning of the therapeutic procedure- multifocal HCC?
In Table 1 bone metastases were observed in 266 patients (after propensity-score matching) and in Figure 3 you presented 302 patients - why such a difference?
Please, include in the discussion informations about the results of other authors' research on similar therapeutic procedures, not only from the Asian regions.
Author Response
Congratulations on the collection so big material. My notes are as follows: Study group comes from Taiwan. This region differs in the etiology of cancer in relation to other parts of the world. There is no information about possible etiology in the manuscript. Only in Table 1, alcoholic liver diseases cases are shown. There is no data about HBV or HCV infections and antiviral therapy during cancer treatment. In so big Taiwan Database, such information should exist, especially that in Figure 3, you put the etiology of HBV or non-HBV.
[Authors’ Reply] We thank the reviewer for highlighting this aspect. We are grateful to be able to respond to the reviewer’s comment here. Actually, Figure 3 has clearly demonstrated that the proportion of HBV infection accounts for 34.7% in this post-propensity score matched cohort. HCV infection, uninfected HCC, and alcoholism account for a total of 65.3%. We did not use etiology to perform propensity score matching simply because severe patient number attrition would occur if we did that. In Figure 3, subgroup analysis has demonstrated that whatever the etiology, either HBV-related or non-HBV-related, achieved the similar benefit favoring the sorafenib plus TACE treatment with a very similar point estimate hazard ratio at 0.76 and 0.75. There is no evidence or rationale from previous publications that prior antiviral therapy will predict the response to sorafenib alone vs. sorafenib + TACE.
In Table 4 presented data concern patients with sorafenib and TACE treatment. A selected group of patients who received two or more TACE treatment is presented. I could not find out if the next TACE was done because of HCC progression or was planned from the beginning of the therapeutic procedure- multifocal HCC?
[Authors’ Reply] We thank the reviewer for the attention to details and having a close look at the results we highlighted in Table 4. We agree with the reviewer regarding the exact purpose of the next TACE. The research purpose of this work is to investigate if add-on TACE to sorafenib for a propensity score-matched ‘homogeneous’ population of patients with advanced HCC having Child A liver reserve after considering immortal time bias, would have a survival benefit over sorafenib alone treatment arm. Table 4 is to ask if just one session of TACE would show the same trend of benefit as more than one TACE or not.
Moreover, our analysis did disclose that even one session of TACE still delivers the same direction of risk reduction. This study did not ask if the intervention radiologist was implementing the next TACE for the same or a different multifocal HCC. It would be of no relevance to our research question.
In Table 1, bone metastases were observed in 266 patients (after propensity-score matching), and in Figure 3, you presented 302 patients - why such a difference?
[Authors’ Reply] We are very much grateful to the reviewer for spotting the difference between Table 1 and Figure 3. Indeed, there appears a typo in Figure 3. In the revised Figure 3, we have corrected the number to 266 after checking our data.
Please, include in the discussion information about the results of other authors' research on similar therapeutic procedures, not only from the Asian regions.
[Authors’ Reply] Again, we thank the reviewer for this constructive comment. However, to our knowledge and our re-conduction of literature search updating to the current moment, at this writing, there is no non-Asian data of comparative trials comparing sorafenib alone vs. sorafenib + add-on TACE for patients with advanced stage HCC. That is why we think our paper will contribute to the gap left behind by the lack of studies previously.
Reviewer 2 Report
The manuscript gave a very interesting retrospective analysis on the big clinic data, and got some exciting results. However, the conclusion might still need further consideration after subgrouping the patients data more in detail. So the paper might need more analysis before giving different conclusion.
Firstly, there are some writing mistakes in some place, such as TACE in the figure 1, not TAE.
The figure 2 is kind of confuse.
Since there have a huge difference on the patients numbers of combination therapy of sorafenib and TACE and sorafenib alone treatment. The conclusion based on these clinic data might also induce some unjust results.
About table 1, did the author analyzed the blood serum test, the virus infection, and non-alcohol fatty liver disease related HCC events in all the patience. And all these risk factors to HCC might show influences on the therapy outcome independent on combination therapy or sorafenib alone therapy . though the author showed the subgroup analysis in figure 3. Meanwhile, if the author could compare the various outcome in combination and sorafenib treatment group separately before combing all the patients data together, through the analysis might give some reasonable reason for the conclusion made in the paper different to previous clinic study.
So if the author could divide the patient samples into various subgroups, and then compared the outcome would be better.
Author Response
The manuscript gave a very interesting retrospective analysis of the big clinic data and got some exciting results. However, the conclusion might still need further consideration after subgrouping the patient's data more in detail. So the paper might need more analysis before giving a different conclusion.
[Authors’ Reply] We thank the reviewer for the encouraging comments. In this real-world clinical research, we have contemplated and implemented the best way we could think of to avoid the immortal time bias, utilized time-dependent Cox model when our data did not fit the proportionality assumption, performed sensitivity analysis to see if number session of additional TACE would change the direction of risk of death or not. With the available variables at hand, we have explored adequate investigations to the analysis. We are confident that our results will be reproducible if the same research question is to be investigated in a similar setting.
Firstly, there are some writing mistakes in some place, such as TACE in Table 1, not TAE.
[Authors’ Reply] We do appreciate the reviewer very much for the attention to details and helping us discover a typo in Table 1. In the revised manuscript, we have corrected all the typos.
Figure 2 is kind of confusing. Since there was a huge difference in the patients numbers of combination therapy of sorafenib and TACE and sorafenib alone treatment. The conclusion based on these clinic data might also induce some unjust results.
[Authors’ Reply] We thank the reviewer for the comment. This research resorted to a 1-to-multiple greedy matching algorithm to expand the accrual of eligible patients so that the power would be even more robust. It is like 1:4 randomization in controlled trials although usually, we encounter 1 to 2 randomization more commonly. One-to-multiple matching is frequently employed to increase precision in cohort studies. Please refer to the following citation:
Ref.: Rassen et al. One-to-many propensity score matching in cohort studies. Pharmacoepidemiology and drug safety 2012;21(S2): 69–80. The authors concluded that 1-to-multiple matching can be used to increase precision in cohort studies.
About table 1, did the author analyzed the blood serum test, the virus infection, and non-alcohol fatty liver disease-related HCC events in all the patients. And all these risk factors to HCC might show influences on the therapy outcome independent on combination therapy or sorafenib alone therapy, though the author showed the subgroup analysis in figure 3. Meanwhile, if the author could compare the various outcome in combination and sorafenib treatment group separately before combing all the patients data together, through the analysis might give some reasonable reason for the conclusion made in the paper different to previous clinic study. So if the author could divide the patient samples into various subgroups, and then compared the outcome would be better.
[Authors’ Reply] Again, we greatly appreciate the reviewer for this comment. We are grateful to be able to respond to the reviewer’s comment here. Figure 3 has clearly demonstrated that the proportion of HBV infection accounts for 34.7% in this post-propensity score matched cohort. HCV infection-related, uninfected HCC, alcoholism-related, and non-alcohol fatty liver disease-related HCC account for a total of 65.3%. We did not use etiology as a variable to perform propensity score matching simply because severe patient number attrition would occur if we did that. In Figure 3, subgroup analysis has clearly demonstrated that whatever the etiology, either HBV-related or non-HBV-related, achieved the similar benefit favoring the sorafenib plus TACE treatment with a very similar point estimate hazard ratio at 0.76 and 0.75. There is no evidence or rationale from previous publications that the etiologic risk factor will predict the response to sorafenib alone vs. sorafenib + TACE. The All Cancer Dataset derived from the mother NHI Research Database does not contain laboratory test results. The research question we asked, rendering a hypothesis that is adding sorafenib to TACE could provide better survival outcome when compared with sorafenib alone in a very specific subgroup of patients with Child-Pugh class A advanced HCC. The propensity score-matching cohort study design could be considered as a quasi-experimental design simulating randomized controlled trial.
Reviewer 3 Report
Cancers-540020
KOK et al.
The authors present a clear comprehensive study on TACE + Sorafenib in advanced HCC. The design and the number of patients analyzed are relevant as well as the statistics and the presentation of the data. The language is comprehensive.
Major comment 1
Some statistics might be further clarified to help the reader. To be able to better interpret the Cox model, the authors might provide additional informations on the overall quality of the model. Please include the R2, the concordance, the Likelihood test, the Wald test, and the LogRank test.
Major comment 2
Cox model is interesting but difficult to interpret and transfer into the clinical practice. To help the readers to interpret the data and provide some predictive insight into the study, the authors might provide Positive and Negative Predictive Values (NPV, PPV) for response to TACE +Sorafenib. If this should be the treatment option the clinicians might want to better target the therapy. Thus it would be interesting to provide data on the patients that respond better to this therapy. For this the group “TACE+Sorafenib” might be subdivided into two equal groups size based on the survival time. Then, a multivariate logistic regression might be run to identify the parameters linked and predicting a better response. Finally, NPV and PPV might be then calculated to provide practical tools for clinicians, easier to interpret.
Minor comment 1
In the patient characteristics as well as the cox model, the authors could include if possible the Hepatitis status.
Author Response
The authors present a clear, comprehensive study on TACE + Sorafenib in advanced HCC. The design and the number of patients analyzed are relevant as well as the statistics and the presentation of the data. The language is comprehensive.
[Authors’ Reply] We thank the reviewer for this kind, warm and encouraging comment. We will certainly keep it up in revising our paper.
Major comment 1
Some statistics might be further clarified to help the reader. To be able to better interpret the Cox model, the authors might provide additional information on the overall quality of the model. Please include the R2, the concordance, the Likelihood test, the Wald test, and the log-rank test.
[Authors’ Reply] We thank the reviewer for the suggestion. On the contrary, in our opinion, clinical oncologists, radiologists, and hepatologists would feel uncomfortable and are mostly unaware of the clinical significance and interpretation of R-square, the Likelihood test and Wald test for survival data. We use the hazard ratio with 95% confidence interval plus P-value for Table 2 and Chi-square test from the Mantel-Byar test with P-value in Figure 2. We adopted Mantel-Byar test for comparison of survival data with a time-dependent covariate. We think the result presentation in this paper is the appropriate way for our analysis.
Major comment 2
Cox model is interesting but difficult to interpret and transfer into the clinical practice. To help the readers to interpret the data and provide some predictive insight into the study, the authors might provide Positive and Negative Predictive Values (NPV, PPV) for a response to TACE +Sorafenib. If this should be the treatment option, the clinicians might want to better target the therapy. Thus it would be interesting to provide data on the patients that respond better to this therapy. For this, the group “TACE+Sorafenib” might be subdivided into two equal groups size based on the survival time. Then, a multivariate logistic regression might be run to identify the parameters linked and predicting a better response. Finally, NPV and PPV might be then calculated to provide practical tools for clinicians, easier to interpret.
[Authors’ Reply] Again, we thank the reviewer for the suggestion. Our analysis is for ‘therapeutic’ rather than ‘diagnostic’ study. Our biostatistics expert does not recommend using PPV and NPV to present our results. As to the second part of the comment, this proposed methodology was not planned in our research. We would consider it a new research question after this paper.
Minor comment 1
In the patient characteristics as well as the cox model, the authors could include if possible the Hepatitis status.
[Authors’ Reply] Again, we greatly appreciate the reviewer for this comment. We are grateful to be able to respond to the reviewer’s comment here. Figure 3 has clearly demonstrated that the proportion of HBV infection accounts for 34.7% in this post-propensity score matched cohort. HCV infection-related, uninfected HCC, alcoholism-related, and non-alcohol fatty liver disease-related HCC account for a total of 65.3%. In Figure 3, subgroup analysis has clearly demonstrated that whatever the etiology for subsequent HCC development, either HBV-related or non-HBV-related, achieved the similar benefit favoring the sorafenib plus TACE treatment with a very similar point estimate hazard ratio at 0.76 and 0.75. There is no evidence or rationale from previous publications that the etiologic risk factor will predict the response to sorafenib alone vs. sorafenib + TACE. Thus, we retain our initial methodology plan to indicate the etiology only in the subgroup analysis.
Round 2
Reviewer 3 Report
No further comment for my side.